# The Effects of Nutrition on Linear Growth

**DOI:** 10.3390/nu14091752

**Published:** 2022-04-22

**Authors:** Elena Inzaghi, Valentina Pampanini, Annalisa Deodati, Stefano Cianfarani

**Affiliations:** 1Pediatric Emergency Department and General Pediatrics, ‘Bambino Gesù’ Children’s Hospital, IRCCS, 00164 Rome, Italy; elena.inzaghi@opbg.net; 2Diabetology and Growth Disorders Unit, ‘Bambino Gesù’ Children’s Hospital, IRCCS, 00164 Rome, Italy; valentina.pampanini@opbg.net (V.P.); annalisa.deodati@opbg.net (A.D.); 3Department of Women’s and Children’s Health, Karolinska Institutet, 17165 Stockholm, Sweden; 4Department of Systems Medicine, University of Rome Tor Vergata, 00133 Rome, Italy

**Keywords:** nutrition, growth, children

## Abstract

Linear growth is a complex process and is considered one of the best indicators of children’s well-being and health. Genetics, epigenetics and environment (mainly stress and availability of nutrients) are the main regulators of growth. Nutrition exerts its effects on growth throughout the course of life with different, not completely understood mechanisms. Cells have a sophisticated sensing system, which allows growth processes to occur in the presence of an adequate nutrient availability. Most of the nutritional influence on growth is mediated by hormonal signals, in turn sensitive to nutritional cues. Both macro- and micro-nutrients are required for normal growth, as demonstrated by the impairment of growth occurring when their intake is insufficient. Clinical conditions characterized by abnormal nutritional status, including obesity and eating disorders, are associated with alterations of growth pattern, confirming the tight link between growth and nutrition. The precise molecular mechanisms connecting nutrition to linear growth are far from being fully understood and further studies are required. A better understanding of the interplay between nutrients and the endocrine system will allow one to develop more appropriate and effective nutritional interventions for optimizing child growth.

## 1. Introduction

Linear growth is recognized as a reliable indicator of a child’s general health. Growth pattern varies during life, being particularly fast during fetal life and the first two years of life, then slowing during childhood until puberty, when growth spurt occurs [1]. 

Though growth potential is genetically determined, growth pattern is deeply influenced by endocrine and environmental factors, including psychosocial distress and nutrient availability. These factors act in an extremely sophisticated interplay, differentially intervening during the various phases of growth. Nutrition seems to be particularly relevant during fetal life and the first year of postnatal life, whereas the endocrine control becomes predominant during childhood and puberty. 

Nutrition in early life has not only an immediate effect on growth but also affects future health. Undernourished fetuses and infants are more likely to be short adults, to have increased cardiometabolic risk in adulthood, to give birth to smaller infants, to have lower educational achievement and to experience a lower economic status in adulthood [2,3].

Although nutritional control of growth is predominant during the first phases of life, nutrition remains an important regulator of growth even during childhood and adolescence [1]. In low-income countries, approximately 25% of children younger than five years, present linear growth restriction due to malnutrition [4,5]. Mechanisms regulating weight and linear growth appear to be interconnected, as supported by the observation that in undernourished children linear catch-up growth occurs when 85% of weight for height has been recovered [6].

Though the causal link between nutrition and growth failure has been questioned [7,8], with some studies reporting beneficial effects of nutritional interventions on linear growth, and others showing poor or no effectiveness [9], overall the available data support the concept that nutrition plays a major role in linear growth. 

The effects of macronutrients (proteins, fats and carbohydrates), representing the immediate “building blocks” for growth, are easily conceivable. Nevertheless, optimal growth requires an adequate intake of micronutrients, whose deficiency may contribute to growth retardation. 

Longitudinal bone growth is the result of a complex interplay between several endocrine, paracrine and autocrine factors that act both directly on chondrocytes in the growth plate and indirectly by modulating other factors of the growth-promoting molecular network. Growth hormone (GH), insulin-like growth factor I (IGF-I), sex steroid hormones, thyroid hormones, insulin, leptin and glucocorticoids are key endocrine regulators of growth that can be influenced by nutritional status. The target organ of all these signals is the growth plate, a thin layer of cartilage entrapped between epiphyseal and metaphyseal bone, where all the factors either promoting or inhibiting growth interact with each other, regulating the longitudinal growth of long bones, and ultimately linear growth. 

Abnormal growth patterns occur in different clinical contexts characterized by altered nutritional status. Therefore, a better understanding of the mechanisms connecting nutrition to growth may lead to the development of strategies for optimizing nutritional status and allowing for the recovery of a normal growth pattern.

## 2. Endocrinological Regulators

### 2.1. GH Axis

GH and its main effector, IGF-I, are recognized to be the main regulators of linear growth, by acting mainly but not exclusively in the growth plate. 

GH regulation is sensitive to different nutritional cues, such as glucose. GH secretion is inhibited by glucose load [10], an effect that may be mediated by ghrelin [11], whereas hypoglycemia stimulates GH release [12].

GH has a lipolytic action [13] and influences the distribution of adipose tissue. On the other hand, GH secretion is affected by lipids. Animal models of exposure to high-fat diet showed impairment of GH synthesis and decreased circulating GH levels, likely through the activation of endoplasmic reticulum stress [14]. Other nutrients, which presumably influence the GH axis, include vitamins and microelements [15].

Short fasting stimulates GH secretion, coherently with the lipolytic and hyperglycemic properties of GH [15]. By contrast, prolonged fasting induces peripheral GH resistance [16]. Data from animal studies have shown that inadequate caloric intake inhibits longitudinal bone growth. In male rabbits undergoing 48 h fasting, a significant reduction in the number of both proliferative and hypertrophic chondrocytes was observed [17]. Despite increased GH levels, the hepatic expression of IGF-I was significantly down-regulated and circulating IGF-I was significantly reduced compared with fed controls. These results suggest that the inhibition of longitudinal bone growth and the associated structural changes observed in the growth plate during fasting may be secondary to the low levels of circulating IGF-I. The reduced expression of IGF-I in liver despite increased GH levels, indicates a status of GH resistance induced by fasting. 

GH resistance has been described in different forms of undernutrition, such as decreased total energy intake, isolated protein calorie malnutrition and isolated micronutrient deficiencies [16]. 

An adequate availability of nutrients is required for the anabolic actions of IGF-I, including amino acid uptake into skeletal muscle, peripheral glucose uptake, increased protein synthesis and reduced proteolysis [18]. In case of starvation, the consequent decrease in IGF-I is associated with protein catabolism, which increases amino acids availability for gluconeogenesis, thus favoring adequate levels of glucose for vital organs such as the brain. 

Dysregulation of IGF-I levels occurs in both under and over-nutrition [18], with serum concentration decreasing in response to malnutrition [19]. Lower IGF-I levels during dietary restriction may depend on the state of hepatic GH resistance, driven by alterations in the GH receptor signaling [18]. The degree of dietary restriction has a different effect on GH receptors. A receptor defect would be responsible for reduced IGF-I levels in more severe forms, whereas a post-receptor defect may be involved in less severe forms of malnutrition. 

IGF-I is sensitive to both protein and total energy intake. An adequate intake of both protein and energy is required to normalize IGF-I levels [20]. Dietary essential amino acid intake is important for IGF-I restoration after fasting [21]. Notably, other macronutrients, such as fat, influence IGF-I levels [22], but at a lower degree than protein or total energy. 

In the growth plates of food-restricted mice, decreased IGF-I levels and lower GHR expression have been found [23] and may explain the reduced response to GH administration observed in malnourished animals and children.

Further components of the GH/IGF-I axis are the IGF-binding proteins (IGFBPs), whose main role is to transport IGFs, thereby regulating their availability for peripheral tissues. IGFBPs represent additional mechanisms by which nutrition influences IGF-I concentrations. Indeed, in case of malnutrition, a significant increase in IGFBP-1 and IGFBP-2 levels occurs, thus increasing IGF-I clearance and reducing its bioavailability [24]. IGFBP-3 levels parallel those of IGF-I in malnutrition and may be used as an additional marker of nutritional status [25,26]. 

### 2.2. FGF21

Fibroblast growth factors (FGFs) are a family of proteins that regulate different biological processes, including growth and development. FGF21 is an endocrine factor primarily produced by the liver and adipocytes that acts as a signal of protein restriction. FGF21 regulates metabolism and growth during periods of reduced protein intake and contributes to the adaptation to fasting by stimulation of gluconeogenesis, fatty acid oxidation, and ketogenesis [27,28,29]. In humans, both fasting and protein deprivation are associated with increased FGF21 levels [29,30,31]. 

FGF21 may mediate GH resistance induced by malnutrition, thus contributing to the consequent impaired skeletal growth [27]. The chronic exposure to FGF21 is associated with reduced expression of hepatic GH receptors, inhibition of GH signaling and disruption of GH action in the growth plate [32]. In animals, increased expression of FGF21 during chronic food restriction is associated with reduced bone growth [33]. Notably, growth failure induced by undernutrition is attenuated in FGF21-knockout mice compared to controls [27]. Elevated FGF21 levels are associated with impaired linear growth in very preterm infants, and in primary human chondrocytes FGF21 inhibits GH action on chondrocytes [34]. Furthermore, plasma levels of FGF21 are inversely related to linear growth in infancy [35]. 

### 2.3. Insulin

Insulin is a peptide hormone that binds to membrane-bound receptors in target cells to orchestrate an integrated anabolic response to nutrient availability. Beyond its fundamental metabolic actions, insulin is a potent mitogen, exerting its growth-promoting effects mainly by binding to the IGF-I receptor. Insulin induces chondrocyte differentiation and maturation, and the administration of insulin in hypophysectomized rats stimulates tibial growth [36]. 

Abnormal insulin secretion is associated with alterations of growth. Pancreatic agenesis is associated with intrauterine growth restriction [37], which also occurs in patients with insulin receptor gene mutations [38]. The impairment of growth observed in children with poorly controlled type 1 diabetes depends, in part, on low insulin levels. 

The insulin growth-promoting action is exerted directly or, indirectly, through the regulation of IGF-I release [39]. Insulin signaling induces IGF-I independent actions on chondrocytes, stimulating them to proliferate, differentiate and achieve their final size [40].

### 2.4. Leptin

Leptin is a hormone mainly but not exclusively secreted by white adipose cells. It regulates sense of satiety and metabolism but also acts as a mediator of nutritional effects on growth [41]. Leptin stimulates GH secretion by acting on the hypothalamus [41,42] and, interestingly, exerts a direct peripheral growth-promoting effect in the growth plate by stimulating chondrocyte proliferation and differentiation [43,44]. Consistently, the administration of leptin to Ob/Ob mice reverses metabolic abnormalities and increases femoral length [45,46]. Furthermore, this animal model of leptin deficiency is characterized by reduced GH circulating levels [47], a finding also observed in patients with leptin receptor mutations [47]. Notably, leptin administration to rats with intrauterine growth retardation accelerates the elongation of bones [48]. In humans, however, the growth-promoting effect of leptin is less clear, as leptin mutation has been described in a family of tall subjects [49].

Specific hypothalamic areas are the target of hormones such as leptin and insulin, which provide information regarding nutrient availability, and connect nutritional status to linear growth and the onset of puberty. Melanocortin-3-receptor (MC3R) is a melanocortin receptor, mainly expressed in the brain, whose lack in animals impairs linear growth. Genetic variants of MC3R are associated with adult height in humans [50]. In humans, MC3R deficiency is associated with delayed puberty, impaired growth, reduced adult height and decreased IGF-I levels [51]. MC3R may thus represent a pivotal mediator between nutritional status and linear growth. 

### 2.5. Thyroid Hormone

Thyroid hormone secretion is deeply influenced by nutrition, requiring iodine as a key component, and being also affected by other micronutrients such as selenium, zinc, iron, and vitamin A [52].

The thyroid hormone plays a well-recognized role in regulating growth and skeletal development from the late fetal life to the onset of puberty, as confirmed by growth alterations occurring in case of either excess or deficiency [1]. 

Thyroid hormone influences endochondral ossification, by regulating chondrocyte maturation as well as cartilage matrix synthesis, mineralization, and degradation both directly and indirectly through GH-mediated effects [24,53]. 

## 3. Nutritional Regulators

### 3.1. Macronutrients

Protein and amino acids are recognized as the main nutrients involved in linear growth. Proteins play a permissive role in growth, since they fulfill the metabolic demand of amino acids, required for tissue growth, and increase levels of hormones, such as insulin and IGF-I, which stimulate endochondral ossification. Amino acids are critical for normal growth and matrix formation by chondrocytes [54].

In humans, protein deficiency leads to growth failure [55]. By exposing animals to protein deficiency, tibial linear growth is quickly negatively affected. This growth-limiting effect is neutralized by increasing dietary protein concentrations [56]. 

By contrast, a high protein intake in infancy and early childhood leads to increased growth and higher BMI in childhood [57]. In infancy, breastfeeding has been associated with a slower growth rate [58], but results are not conclusive [59,60]. 

Leucine is a ubiquitous amino acid particularly present in milk and some cereals [61]. Leucine regulates insulin metabolism and exerts anabolic and anticatabolic actions. Leucine stimulates growth through the activation of the mTOR signaling pathway. This pathway integrates different environmental cues to regulate cell growth and homeostasis [62]. mTOR is a serine/threonine protein kinase that belongs to the phosphoinositide 3-kinase (PI3K)-related kinase family and interacts with several proteins to form two distinct complexes, named mTOR complex 1 (mTORC1) and 2 (mTORC2). mTORC1 upstream signals include amino acids (especially leucine and arginine), stress, oxygen, energy, and growth factors [63]. mTORC1 favors cell growth by promoting anabolic processes such as protein and lipid synthesis and by simultaneously inhibiting autophagy. Moreover, activated mTOR stimulates angiogenesis, which allows nutrients to reach the cells [47] and influences osteoblast differentiation [64]. mTOR signaling stimulates chondrocyte differentiation [65] and affects chondrocyte autophagy in the growth plate [47]. Notably, mTORC1 signaling is active in the hypothalamus, where it integrates signals from circulating nutrients (glucose, amino acids, lipids) and hormones (leptin, insulin) to synchronize energy balance and growth. Intracerebroventricular administration of leucine and leptin promotes mTORC1 activity and reduces food intake in rats. 

### 3.2. Micronutrients

The effects of single or micronutrient mixture supplementation on linear growth have been investigated in different studies, which have yielded conflicting results. This inconsistency may depend on the extreme variability of nutritional interventions as well as differences in control groups and study cohorts. It has to be pointed out that malnourished children have multiple nutrient deficiencies that affect the efficacy of single supplementations. 

Zinc is a central component of hundreds of enzymes involved in cell growth and differentiation as well as immune function. The first evidence of zinc involvement in growth derived from the observation that human zinc deficiency secondary to acrodermatitis enteropathica, an inborn metabolic error causing reduced intestinal absorption of zinc, was associated with impaired growth and increased susceptibility to infections [66]. A growth-promoting effect of zinc supplementation has been observed in animals [67]. For instance, in rats, zinc deficiency induces structural changes of the growth plate and reduces the length of tibias and femurs [47,68]. Zinc may also influence the growth plate by reducing IGF-I secretion as well as peripheral actions of IGF-I [69]. 

Despite zinc supplementation effects on growth being extensively studied, results in humans are inconsistent, partially due to the considerable variability in inclusion criteria. 

A small effect of zinc supplementation in stimulating growth in pre-pubertal children was reported [70], a finding consistent with results obtained in children under 5 years of age in developing countries [71]. Overall, the zinc positive effect on linear growth seems to be particularly significant after 2 years of age [72]. However, other studies reported conflicting results [73,74,75]. A systematic review of studies reporting data from a total of more than 27.000 children from low- and middle-income countries, under 5 years of age, showed that zinc supplementation has little or no effect on anthropometric indices [76]. 

The exact mechanism by which zinc influences linear growth is still unclear. It has been suggested that an adequate zinc intake is needed for chondrogenesis, collagen synthesis, osteoblast function, and calcification of bone [77].

Vitamin D influences endochondral ossification by stimulating cellular maturation through the vitamin D receptor [47]. The vitamin D receptor (VDR) is a member of the nuclear receptor superfamily and regulates the expression of numerous genes involved in calcium/phosphate homeostasis, cellular proliferation and differentiation, and immune response, largely in a ligand-dependent manner.

VDR is largely expressed in chondrocytes. In the human fetal growth plate, vitamin D promotes chondrocyte differentiation by stimulating the expression of IGF-I and GH receptor genes [24,78]. 

A recent extensive meta-analysis aimed at evaluating the effects of vitamin D supplementation on several clinical outcomes in children under five years of age found little or no effect of vitamin D on linear growth [79].

Calcium homeostasis is essential for bone health and growth. In animals, calcium deficiency causes reduced bone mineralization and reduced bone strength without affecting linear growth [80]. Vitamin D and calcium administration restore normal bone growth in children with nutritional rickets [81]. Low intake of calcium and vitamin D, likely due to inadequate milk intake after weaning, may favor stunting in African children [82]. In adolescent boys (aged 16–18), 13 months of calcium supplementation was associated with increased height [83].

Vitamin A and its derivative, retinoic acid, have no clear effects on growth [84]. Trials based on vitamin A supplementation have reported little or no benefit on linear growth [73,74]. By contrast, according to a recent extensive meta-analysis including five studies assessing the effect of vitamin A on linear growth in children, vitamin A supplementation may exert a positive effect on linear growth in children older than 2 years [77].

Iron supplementation was reported to stimulate growth only in children with iron deficiency anemia [75]. Consistently, a meta-analysis of randomized controlled trials assessing the effect of iron interventions on the growth of children younger than 5 years showed no significant effects [65]. These results were confirmed by a meta-analysis including 14 studies, performed in low- and middle-income countries on subjects with ages ranging from 34 to 167 months [77]. 

High dietary copper intake promotes growth in pigs [85], whereas in rats its deficiency results in low serum IGF-I levels but high IGF-I in bones [86]. Copper supplementation increases IGF-I and IGFBP-3 concentrations in culture media of chondrocytes, promoting their proliferation. Data about copper supplementation trials in infants and children are scant.

Iodine is an essential component of the thyroid hormone, through which it exerts its main effects on growth. Iodine deficiency affects people of all ages, children and adolescents being the most vulnerable. The widespread salt iodization programs have lowered the risk of iodine deficiency, which is nevertheless still present in many regions [87,88]. Data on the effect of iodine supplementation show no effect of this micronutrient on linear growth [77]. By evaluating a cohort of approximately 300 children followed up to 4 years after the assumption of iodized oil, an improvement of linear growth was observed [89]. 

A multiple-micronutrient approach is more effective on growth than interventions based on single micronutrients [73,90]. A thorough meta-analysis assessing the effects of different nutritional interventions on anthropometric measures (changes in height, weight, and weight-for-height z scores) showed that interventions including iron or vitamin A alone did not influence growth and zinc alone exerted only a small positive effect on weight-for-height z score but no effect on height or weight gain [74]. By contrast, interventions based on multiple micronutrients seem to be effective in stimulating linear growth. The combined administration of vitamin A and zinc in Indonesian short children (aged 48-60 months) without underlying diseases led to increased IGF-I levels and height Z-scores [91]. 

## 4. Clinical Implications

Malnutrition refers to a condition characterized by an imbalance between nutrient requirement and intake. In industrialized countries, this condition can occur in the presence of chronic or acute illnesses but, in the last decades, different forms of malnutrition have emerged in childhood and adolescence. If an excessive energy intake can lead to overweight and obesity, eating disorders, characterized by voluntary inadequate caloric intake and/or purging behaviors, lead to undernutrition. 

### 4.1. Anorexia Nervosa

Anorexia nervosa (AN) is an increasingly widespread disorder with a progressively decreasing age at onset. It affects approximately 0.2-1% of adolescents and young women in developed countries [92], but up to 14% of adolescents are presumed to have “partial syndromes or eating disorder not otherwise specified” with signs and symptoms of malnutrition.

These patients typically have multiple endocrine disorders [92,93], mostly representing mechanisms of adaptation to chronic starvation. Patients with AN usually have acquired GH resistance, with higher GH but low IGF-I levels. Considering that GH stimulates gluconeogenesis, its higher levels represent an adaptive mechanism favoring the maintenance of euglycemia in subjects with a reduced availability of energy substrates. Refeeding normalizes GH secretion in these patients [94]. 

Patients with AN show a persistent impairment of growth rate. Though nutritional interventions induce an increased linear growth, catch-up growth is often incomplete [95]. In these subjects, IGF-I may represent a biomarker of nutritional status, as its changes reflect BMI fluctuations [96]. 

### 4.2. Obesity 

The opposite extreme of malnutrition is represented by an excessive food intake leading to overweight or obesity.

Obese children typically show accelerated linear growth in childhood associated with an accelerated maturation of the epiphyseal growth plate, ultimately leading to an adult height consistent with the genetic potential. Adipose tissue is a source of different hormones, which may affect the linear growth of obese children. Leptin, which is mainly produced in the adipose tissue, informs the hypothalamus about the nutritional status (by binding MC3R), thus activating the kisspeptin system and the GnRH pulse generator, ultimately regulating puberty onset and progression, and GHRH-releasing neurons, ultimately increasing pituitary GH secretion [97]. 

Obesity influences the IGF system. Obese subjects usually have reduced GH levels [98,99], which normalize after weight loss [15], and normal or increased IGF-I levels [100,101].

Ghrelin, which acts as an endogenous GH-releasing factor, is a peptide mainly produced in the stomach and acts as a ligand of the GH secretagogue receptor (GHSR). Ghrelin stimulates food intake, rises in fasting states, decreases after eating and has been proposed as a possible regulator of GH release in obesity [102]. Moreover, ghrelin is also synthesized by chondrocytes in the growth plate, thus exerting a combined central and peripheral growth-promoting action [103].

Insulin, whose levels are increased in obese children, reduces GH secretion [104] and exerts both direct and indirect growth-promoting actions [36]. 

## 5. Conclusions

Nutrition plays a key role in the regulation of growth in fetal life, infancy, childhood and adolescence. Nutrients represent the fundamental bricks for cartilage and bone development and, at the same time, regulate a complex network of hormones and growth factors, which act in an endocrine, paracrine and autocrine fashion, finely controlling growth plate physiology (Figure 1). A better understanding of the interplay between nutrients and the endocrine system will allow one to develop more appropriate and effective nutritional interventions for optimizing child growth. 

## Figures and Tables

**Figure 1 nutrients-14-01752-f001:**
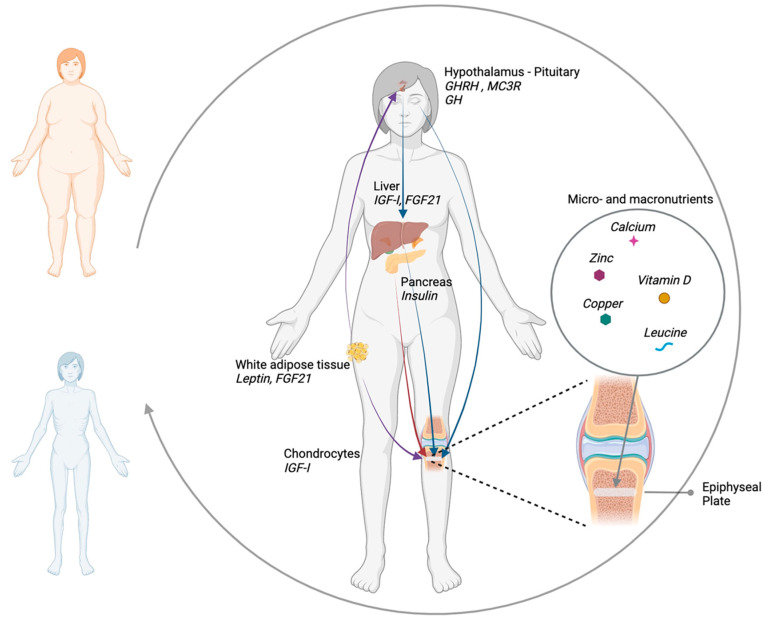
Schematic representation of the interplay between nutritional status and endocrine regulators of growth: growth hormone (GH) resistance induced by prolonged fasting impairs GH direct and insulin-like growth factor I (IGF-I) mediated action on chondrocytes. The increased secretion of fibroblast growth factor 21 (FGF21) by liver and adipose tissue in malnutrition contributes to GH resistance by reducing hepatic GH receptors expression and disrupting GH action in the growth plate. Insulin promotes growth by acting both directly on chondrocytes and indirectly, stimulating IGF-I production. Leptin, produced by adipose tissue, stimulates growth hormone-releasing hormone (GHRH) secretion by the hypothalamus and exerts a direct peripheral growth-promoting effect in the growth plate by stimulating chondrocyte proliferation and differentiation. Hypothalamic melanocortin 3 receptor (MC3R) integrates signals of metabolic status that affect body growth and sexual maturation. Micronutrients such as zinc, copper, calcium, vitamin D and macronutrients such as aminoacids, in particular leucine, exert a direct effect on the epiphyseal growth plate by influencing chondrocyte differentiation and proliferation.

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
