# Peer review of "The Effects of Nutrition on Linear Growth"

_nutrients, 2022, doi:10.3390/nu14091752_

Round 1

Reviewer 1 Report

This review describes nutritional involvement in Linear Growth, but Abstratct isn't clear, and the introduction doesn't tell us why we're about to outline it. This is probably because the authors just listed their knowledge and information, but cannot express what kind of information the reader needs. There is no wrong description for each item, but many of the references are old. The authors need to make sure if they really don't need to update. They also provided only one general  figure, and I would like to ask for further ingenuity such as inserting tables and figures in order to promote the understanding for the reader.

Author Response

This review describes nutritional involvement in Linear Growth, but Abstract isn't clear, and the introduction doesn't tell us why we're about to outline it. This is probably because the authors just listed their knowledge and information, but cannot express what kind of information the reader needs. There is no wrong description for each item, but many of the references are old. The authors need to make sure if they really don't need to update. They also provided only one general figure, and I would like to ask for further ingenuity such as inserting tables and figures in order to promote the understanding for the reader.

The abstract and introduction has been partially restructured and improved according to the reviewer’s suggestion. Additionally, updated references have been added throughout the text.

Reviewer 2 Report

This paper reviewed basic concepts on the relationship of nutrients, endocrine system and linear growth. This ideas are clear and the writing is easy to follow.  However some major concerns are listed here.

  1. The title and the main focus of this review seems to be about the nutritional effect on linear growth, but the effect of nutrients on linear growth only contribute to a smaller part of the review and are not sufficiently described. Especially on vitamin A, D, iodine and iron. The authors could further elaborate what are dosage of the supplements used, the estimated effect size on the outcomes, or how long is the intervention duration, where significant improvements on micronutrient status have been observed? How about the best timing to intervene? These have not been discussed, a more in-depth discussion is needed.

  1. The authors mentioned that ‘Results in humans are inconsistent. A small effect of zinc supplementation in stimulating growth in pre-pubertal children was reported but not confirmed in other studies ’. however, there are quite many trials demonstrating the positive impacts of zinc supplements on promoting linear growth, thus its beneficial effects are quite promising (line 245). There is also more recent evidence on the impacts of zinc supplementation on linear growth from systematic review and meta-analysis The authors may need to update the references accordingly. for example:
  2. Gera T., Shah D. & Sachdev H. S. Zinc supplementation for promoting growth in children under 5 years of age in low- and middle-income countries: a systematic review. Indian Pediatr 2019;56(5):391-406.
  3. Imdad & Bhutta Z. A. Effect of preventive zinc supplementation on linear growth in children under 5 years of age in developing countries: a meta-analysis of studies for input to the lives saved tool. BMC Public Health 2011;11(Suppl 3):S22.

  1. The effect of vitamin D on linear growth ended abruptly at line 250 and continued at line 265.

  1. The author elaborated the regulation of growth by GH axis, FGF21, insulin and leptin. Other hormons such as thyroid is essential for bone maturation, and deficiency in children is often associated with short stature [1]. It should not be missed that thyroid hormones regulate growth andare modulated by nutrients.

  1. In the subheading 6. Macronutrients, only protein and amino acids have been discussed, how about carbohydrate and fat? How about caloric restriction? These have not been discussed as well.

The paragraph on Anorexia nervosa (line 303-306) doesn’t make sense to me. Is hard to catch up.

The authors should discuss the effects of glucose and lipids on endocrine hormones and growth. For example, growth hormone release is suppressed by glucose loading and stimulated by hypoglycemia, and high fat diet inhibits the synthesis and secretion of growth hormone in rats [refs below].

  1. Mirella Hage, Peter Kamenický, Philippe Chanson. Growth Hormone Response to Oral Glucose Load: From Normal to Pathological Conditions. Neuroendocrinology, 2019; 108(3): 244-255.
  2. Roth J, Glick SM, Yalow RS, Bersonsa. Hypoglycemia: a potent stimulus to secretion of growth hormone. Science, 1963 May 31; 140(3570):987-8.
  3. Ying Gong, Jianmei Yang, Shuoshuo Wei. Lipotoxicity suppresses the synthesis of growth hormone in pituitary somatotrophs via endoplasmic reticulum stress. J Cell Mol Med, 2021 Jun; 25(11):5250-5259.

  1. The review could be structured by a number of sections such as mechanism or regulation on linear growth, effects of nutrients on linear growth, rather than numbering all in sequence.

  1. Citations are missing at the first paragraph of 7. Micronutrient (line 222-231)

  1. I like the figure, is nice and clear. But the authors did not mention where the Figure 1 should be inserted.

Author Response

This paper reviewed basic concepts on the relationship of nutrients, endocrine system and linear growth. This ideas are clear and the writing is easy to follow.  However some major concerns are listed here.

 Point 1. The title and the main focus of this review seems to be about the nutritional effect on linear growth, but the effect of nutrients on linear growth only contribute to a smaller part of the review and are not sufficiently described. Especially on vitamin A, D, iodine and iron. The authors could further elaborate what are dosage of the supplements used, the estimated effect size on the outcomes, or how long is the intervention duration, where significant improvements on micronutrient status have been observed? How about the best timing to intervene? These have not been discussed, a more in-depth discussion is needed.

According to the points raised by the reviewer, we have extended the text related to nutrients, in order to provide a more comprehensive and updated explanation about the role of vitamin A, D, iodine and iron on growth. The text has been changed. Nutrition exerts its influence on growth directly and indirectly, with a close interplay with other main actors such as hormones and growth factors. Therefore, we think it’s useful to include these components in the text. Regarding the suggestions to include “dosage of the supplements used, the estimated effect size on the outcomes, or how long is the intervention duration”, in our opinion this information is beyond the aim of the current review. Moreover, the different studies use different protocols (type of treatment, duration and dosages)  and data have been reported in the meta-analyses cited in the text and listed in the references.

Point 2. The authors mentioned that ‘Results in humans are inconsistent. A small effect of zinc supplementation in stimulating growth in pre-pubertal children was reported but not confirmed in other studies ’. however, there are quite many trials demonstrating the positive impacts of zinc supplements on promoting linear growth, thus its beneficial effects are quite promising (line 245). There is also more recent evidence on the impacts of zinc supplementation on linear growth from systematic review and meta-analysis The authors may need to update the references accordingly. for example:

According to the reviewer’s suggestion we have extended the text and included the suggested reference. However, as underlined in the text, the results though promising are still preliminary and often conflicting.

Point 3. The effect of vitamin D on linear growth ended abruptly at line 250 and continued at line 265.

 The text has been modifies accordingly.

Point 4. The author elaborated the regulation of growth by GH axis, FGF21, insulin and leptin. Other hormones such as thyroid is essential for bone maturation, and deficiency in children is often associated with short stature [1]. It should not be missed that thyroid hormones regulate growth and are modulated by nutrients.

According to the reviewer suggestion we have included a paragraph on thyroid hormone and modified the text.

 Point 5. In the subheading 6. Macronutrients, only protein and amino acids have been discussed, how about carbohydrate and fat? How about caloric restriction? These have not been discussed as well.

We understand the reviewer’s point. We focus  on protein and amino acids role in linear growth as the link between these macronutrients and growth has been extensively investigated and play by far the major role in growth regulation. The experimental and clinical evidence linking carbohydrates and lipids to linear growth is scant. 

Point 6. The paragraph on Anorexia nervosa (line 303-306) doesn’t make sense to me. Is hard to catch up.

We think that clinical implications represent a significant aspect to be tackled to translate the experimental evidence into clinical practice. The inclusion of clinical conditions with growth abnormalities, characterized by altered nutritional status may represent useful and real life  examples of the close link between nutrition and growth.

Point 7. The authors should discuss the effects of glucose and lipids on endocrine hormones and growth. For example, growth hormone release is suppressed by glucose loading and stimulated by hypoglycemia, and high fat diet inhibits the synthesis and secretion of growth hormone in rats [refs below].

According to the reviewer’s suggestion, we have included the suggested references and we have modified the text.

Point 8. The review could be structured by a number of sections such as mechanism or regulation on linear growth, effects of nutrients on linear growth, rather than numbering all in sequence.

 According to the reviewer’s suggestion, we have introduced two main paragraphs (named: endocrinological regulators and nutritional regulators).

Point 9. Citations are missing at the first paragraph of 7. Micronutrient (line 222-231)

The first paragraph is an overall introduction of the role of different micronutrients in growth. We have added specific references to each following point.

Point 10. I like the figure, is nice and clear. But the authors did not mention where the Figure 1 should be inserted.

Figure 1 is cited at line 350. The figure is inserted at the end of the paper, as a summarizing figure.

Reviewer 3 Report

Despite the title "nutrients", the majority of contents refer to hormones. Changing the title is recommended.

Author Response

Point 1. Despite the title "nutrients", the majority of contents refer to hormones. Changing the title is recommended.

We understand the reviewer’s point of view. Nevertheless, many of the nutrition effects on growth are mediated by hormones; therefore, we think that it is important to include these aspects in the text. The term “nutrients” in the title is general, including both direct as well as indirect effects.

Round 2

Reviewer 2 Report

Yes the manuscript has been improved.